# Assessing antimicrobial use patterns in Christian Health Association of Malawi (CHAM) health facilities: A cross-sectional study protocol

**Evans Chimayi Chirambo** [1,2,3]*, **Francis Kachidza Chiumia** [3], **Dumisani Nkhoma** [1], **Collins Mitambo** [4], **Sharon Odeo** [5‡], **Patience Khomani** [6‡], **Innocent Chibwe** [4‡], **Simon Matchado** [7‡], **Pacharo Matchere** [1], **Beatrice Chiweza** [1], **Elled Mwenyekonde** [1‡], **Elizabeth Kampira** [3], **Happy Makala** [1]

1 Department of Health Programmes, Christian Health Association of Malawi (CHAM), Lilongwe, Malawi, 2 Malaria Department, Malawi-Liverpool Wellcome Programme (MLW), Blantyre, Malawi, 3 School of Life Sciences and Allied Health Professions, Kamuzu University of Health Sciences (KUHES), Blantyre, Malawi, 4 Department of Public Health Institute of Malawi, Ministry of Health-Antimicrobial Resistance National Coordinating Centre (MOH-AMR-NCC), Lilongwe, Malawi, 5 Department of Health Programmes, Ecumenical Pharmaceutical Network (EPN), Kilimani, Nairobi, Kenya, 6 Department of Pharmacy, Ekwendeni College of Health Sciences (ECOHS), Mzuzu, Mzimba, Malawi, 7 Department of IT, Pharmacy and Medicines Regulatory Authority (PMRA), Lilongwe, Malawi

☯ These authors contributed equally to this work.
‡ SO, PK, IC, SM, and EM also contributed equally to this work.
* ecchirambo@mlw.mw

**Data Availability Statement:** No datasets were generated or analysed during the current study. All

## Abstract

The threat of antimicrobial resistance (AMR) in Malawi is high with reported mortality of 19,300 annually, the 23rd highest age-standardised mortality. One of the drivers of AMR is misuse of antibiotics, a phenomenon that has not been adequately researched in Malawi. This study aims to investigate antimicrobial use using health facility, prescribing and patient indicators in Christian Health Association of Malawi (CHAM) health facilities. This will be a multiple cross-sectional study, which will collect data from facilities (22), prescriptions (660), and patients (303). Data will be collected using KoboToolbox v2021, and exported into Microsoft Excel version 2016 for cleaning and coding. Variables will be categorised according to the antimicrobial use indicators. The study will use STATA Version 14 statistical software for data analysis. Subsequently, facilities will be entered into ArcGIS Version 10.7.1 to map hotspots of irrational antimicrobial use. The study will run from October 2024 to June 2025. This study will provide detailed information on frequently used antimicrobials, the cost of antimicrobials relative to medicine budget, the intensity of exposure to antimicrobials, the availability of antimicrobials, patients' understanding of antimicrobials use, and availability of important documents for antimicrobial use. Secondarily, the study will unravel the prevalence of irrational antimicrobial use, the main factors contributing to it, and location where irrational use is most prevalent. These findings will inform the national antimicrobial stewardship action plan, aiming to safeguard the available antimicrobials.

relevant data from this study will be made available upon study completion.

**Funding:** The author(s) received no specific funding for this work.

**Competing interests:** The authors have declared that no competing interests exist.

# Introduction

Antimicrobial resistance (AMR) has emerged as a global health crisis, posing a significant threat to the effective treatment of infectious diseases. World Health Organisation (WHO) estimates that 1.27 million annual deaths are directly linked to bacterial AMR while 4.97 million annual deaths are associated with bacterial AMR [1]. It is further predicted that the global annual mortality rate will reach 10 million by 2050 if no measures are taken to curb AMR [2]. In Malawi, 3,600 deaths are directly caused by bacterial AMR annually and 15,700 deaths associated with bacterial AMR, the 23rd highest age-standadised mortality rate per 100,000 population associated with AMR across 204 countries [3]. Unfortunately, drivers of AMR such as irrational antimicrobial use are not well elucidated in Malawi.

Christian Health Association of Malawi (CHAM), established in 1966, is a faith-based non-governmental organisation that runs 194 health facilities. These health facilities are categorised as full-fledged hospitals which are similar to a district hospital and offer specialised healthcare services, community hospital which are mid-level hospitals with very few or no specialised healthcare services, and health centres which provide basic healthcare services. The health facilities are distributed in 27 of 28 Malawi's political districts except Mwanza District, and offers up to 30% of the national healthcare services, second only to the State, with more focus on poor people in hard-to-reach areas [4].

CHAM is one of the highest consumers of antimicrobials in Malawi. However, data is lacking on antimicrobial use in CHAM's health facilities. Specifically, no study has looked at antimicrobial use indicators such as health facility, prescribing, and patient data for any of CHAM's health facilities. This study aims to investigate antimicrobial use in CHAM health facilities using health facility, prescribing and patient indicators. Additionally, the study will unravel irrational antimicrobial use in these health facilities. Understanding the patterns of antimicrobial use within CHAM will facilitate the development of targeted interventions to promote appropriate health facility and prescribing practices, and optimise patient outcomes.

# Objectives

## The broad objective

To investigate antimicrobial use using health facility, prescribing and patient indicators.

## The specific objectives

1. To find out the extent of antimicrobial use based on health facility, prescribing and patient indicators.

2. To measure the prevalence of irrational use of antimicrobials using health facility, prescribing, and patient indicators.

3. To identify two factors for each of health facility, prescribing and patient indicators that increase irrational use of antimicrobials.

4. To map-out health facilities with irrational antimicrobial use (irrational use above WHO set percentage of irrational use).

## Methods

### Study aim

This study seeks to identify shortcomings and areas for improvement in antimicrobial use practices. The insights gained from this analysis will be invaluable in guiding the development and implementation of targeted interventions aimed at optimising antimicrobial usage, combating AMR, and ultimately enhancing patient care outcomes. Additionally, the study will provide a basis for formulation of key policies and intensification of regulations in antimicrobial use.

### Study design

This will be a multiple cross-sectional study design. The study will collect primary data on health facility, prescribing and patient indicators as per inclusion and exclusion criteria.

### Study setting and sites

The study will collect data from CHAM health facilities. These health facilities are located in 27 of 28 political districts of Malawi. These health facilities are evenly distributed across the country, with the majority located at an average distance of over eight kilometres from the nearest health facility, whether CHAM or State-owned. Ninety percent of these health facilities are located in rural and hard-to-reach areas.

The study will be coordinated from CHAM Secretariat in Lilongwe, the Capital City of Malawi. Additionally, data cleaning, analysis and dissemination will be planned and executed from the Secretariat.

### Participants characteristics (sample size and selection)

The study will collect data on health facility, prescribing and patient indicators of antimicrobial use. For sample sizes of health facility and prescribing data, WHO recommends 20 health facilities and at least 600 prescriptions for a cross-sectional study to provide sufficient data to detect differences in antimicrobial use [5]. In this study, 10% will be added to both the health facility and prescribing data's samples sizes to account for withdraw of consent or incomplete data, resulting in final sample sizes of 22 health facilities and 660 prescriptions (30 prescriptions per health facility). The sample size for patient data, calculated using the cross-sectional study sample size formula by Jaykaran Charan and Tamoghna Biswas [6], is 275 patients. This calculation assumes standard normal variate (Z) of 1.96, expected proportion from previous study of 0.763 [7] and precision of 5%. Adding 10% to account for withdraw of consent or incomplete data, the final patient data's sample size for this study is 303.

For the 22 health facilities, we used Epitools [8], a random number generator, to randomly obtain the required numbers for the health facilities' sample selection. The generator was set with the sample size of 22 (health facilities), a minimum value of 1, and a maximum value of 175, which represents the total number of facilities that admit patients within the CHAM network of health facilities out of 194 health facilities. The generated numbers and selected health facilities are as shown in Fig 1, Table 1, and S1 File. For prescribing data, simple random sampling will be used to select 30 prescriptions per health facility from the 22 health facilities. If any of the selected health facilities is closed or inaccessible by any means of transport or communication, it will be replaced by another health facility selected at random. Patient data will then be selected from the 660 prescriptions, with each health facility contributing 13 random prescriptions, totaling 286 patient data. The remaining 17 patient data points will be randomly selected from the remaining prescribing data to get the 303 patient data.

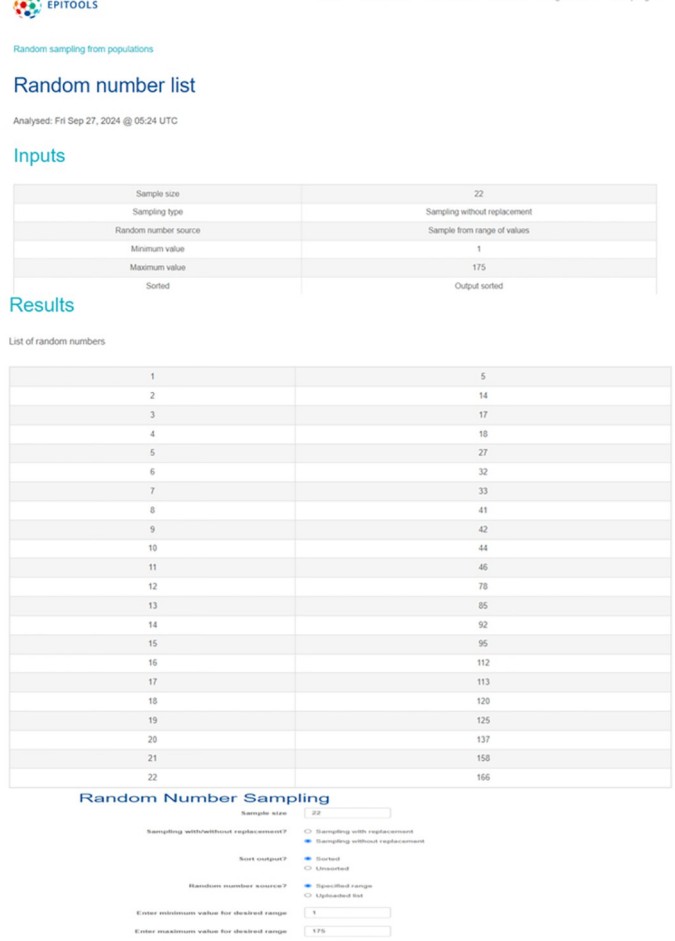

**Fig 1. Randomly generated numbers and selected health facilities.**

## Data collection

Two trained personnel will collect data for each health facility; pharmacy profession will collect health facility data while a clinician will collect prescribing and patient data. The study will use two online questionnaires, one for health facility data and the other for patient and prescribing data, on KoboToolbox v2021.2.4 (S2 File). Health facility data will be the first to collect. This will be followed by simultaneous collection of prescribing and patient data.

The study will collect data from health facilities and admitted patients based on the following inclusion and exclusion criteria.

Inclusion criteria;

1. Patients who have been admitted for up to one month as of the day of data collection.

2. Prescriptions issued within one month prior to the day of data collection.

3. Facilities that are opened and operational on day of data collection.

Exclusion criteria;

1. Children under the age of five

**Table 1. Selected facilities from randomly generated numbers.**

| # in original list | Name of health facility | District | Randomly generated number |
|---|---|---|---|
| | **CENTRAL REGION** | | |
| 5 | St Cynthia chisankhwa HC | Chitipa | 5 |
| 14 | Ekwendeni Hospital | Mzimba | 14 |
| 17 | Padro Pio Mzimba | Mzimba | 17 |
| 18 | St John's Hospital | Mzimba | 18 |
| 27 | Luwazi HC | Nkhatabay | 27 |
| | **CENTRAL REGION** | | |
| 32 | Mtendere Community Hospital | Dedza | 32 |
| 33 | Mua Hospital | Dedza | 33 |
| 41 | St Joseph Chiphwanya HC | Dedza | 41 |
| 42 | Kaundu HC | Dedza | 42 |
| 44 | Madisi Hospital | Dowa | 44 |
| 46 | Nkhamenya Community Hospital | Kasungu | 46 |
| 78 | Ganya HC | Ntcheu | 78 |
| 85 | Nsipe HC | Ntcheu | 85 |
| 92 | Chinthembwe HC | Ntchisi | 92 |
| 95 | Kaphatenga HC | Salima | 95 |
| | **SOUTHERN REGION** | | |
| 112 | Mindati HC | Chikhwawa | 112 |
| 113 | St Joseph Hospital Nguludi | Chiradzulu | 113 |
| 120 | Nthorowa HC | Machinga | 120 |
| 125 | Katema HC | Mangochi | 125 |
| 137 | Billy Riodan HC | Mangochi | 137 |
| 158 | St Joseph Mitengo | Thyolo | 158 |
| 166 | Chilipa HC | Zomba | 166 |

2. Illegibly written prescriptions

3. Incomplete or invalid prescriptions

4. Patient in the intensive care unit (ICU)

5. Facilities that are not reachable by any means of transport or communication.

## Study outcome

Health facility indicators' outcomes include the presence of Standard Treatment Guidelines (STG) and an Essential Medicines List (EML), availability of key antimicrobials at the hospital, the average number of days that key antimicrobials are out of stock, and the percentage of antimicrobial expenditure relative to the total hospital medicines cost.

For prescribing indicators, the study will provide the extent of antimicrobial use through percentage of hospitalisation with one or more antimicrobial used and average number of antimicrobials prescribed per hospitalisation in which antimicrobials were prescribed. Furthermore, the findings will include average cost of antimicrobials prescribed per hospitalisation, average duration of prescribed antimicrobial treatment for intensity of exposure to antimicrobial, percentage of antimicrobials prescribed consistent with the hospital formulary list and by generic name to measure the degree of conformity to national prescribing policies, and average number of drug encounter to measure the degree of polypharmacy. The patient indicators'

outcomes include percentage of prescribed antimicrobials that are actually dispensed and administered, average consultation and dispensing time, the percentage of medicines adequately labelled, and the percentage of patients receiving correct doses of medicines.

Overall, the study will unravel the prevalence and drivers of irrational antimicrobial use in CHAM health facilities. Additionally, the study will also provide insights into the geographical distribution of irrational antimicrobial use within these health facilities.

## Data management plan

This data from KoboToolbox v2021.2.4 will be downloaded to Microsoft Excel Version 2016, cleaned and coded. The data will be stored on a password-protected computer, and access will be restricted to only study team, and relevant authority under full authorisation of the PI. The data will not be connected to patients as numbers and letters will be used to identify participants. Data will be archived for any future research and participants will assent and consent to the future use of data for research purposes.

## Safety plan

This is an observational study hence there are no direct safety concerns related to the study approach. However, data collecting team will call upon clinical team to attend to any patient requiring clinical attention while responding to the study. Additionally, personal protective equipment (PPE) such as latex examination gloves, apron, face masks and hand sanitisers will be provided to personnel collecting prescribing and patient data to prevent exposure to infectious agents. The data collecting team will also be encouraged to wear lab coats or scrub suits.

## Data type and statistical analysis

Statistical analysis will be performed using STATA SE version 14. The data analysis will include frequencies, mean, standard deviation (SD) and percentage to describe data distribution and spread. Additionally, the study will use one-way analysis of variance (ANOVA) for numerical data and Pearson's chi-squared or fisher's exact (depending on the cell number) tests for categorical data, to check for association among different variables. P-value less than 0.05 will be considered statistically significant at 95% confidence interval for all data.

## Ethical considerations and declarations

The protocol was approved by College of Medicine Research Ethics Committee (COMREC) on 15[th] May 2024, protocol number P.04/24-0651. Health facilities, prescriptions and patients will be identified by numbers. Consent will be sought from adults while assent will be obtained for children. Participation in the study will be voluntary. Additionally, the study team secured approval from CHAM to implement the study in the organisation's health facilities and this communication will be sent to health facilities before data collection. The data collection team will obtain authorisation from the health facility in-charge before visiting the wards during data collection. Data collected for the study will be used for research purposes only and nothing else.

## Study status and timelines

The study will run from January 2024 to June 2025. Currently, the implementation is at preparatory stage that will be followed by pre-testing the questionnaires then data collection up-to data dissemination (Fig 2).

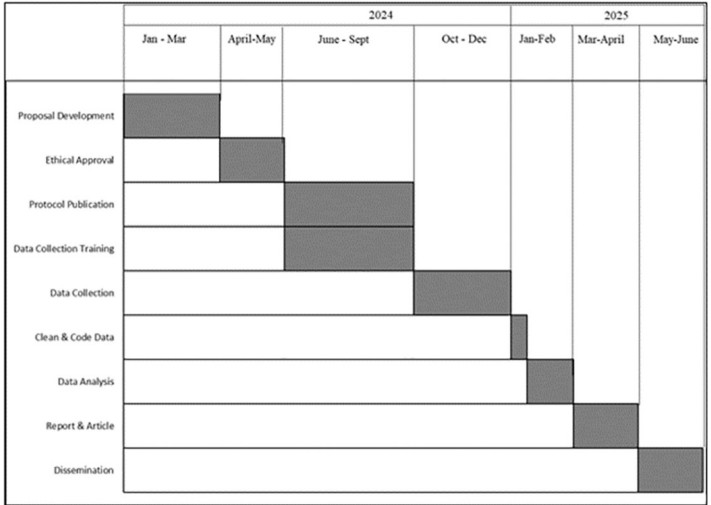

**Fig 2. The study activities and timelines.**

## Discussion

### Study limitations and challenges

The study will involve self-reporting and review of documents that might introduce errors because of the knowledge level of the reporter or the quality of documentation, possibly affecting the accuracy and quality of collected data. To address this, multiple data sources including patients, patient file, and health passports will be used. Where data from the reviewed sources is contradictory, the clinician who assessed the patient or has detailed information about the patient will be consulted for further clarification. Additionally, data from patient files and health passports may be incomplete or inaccurate because of unclear handwriting or poorly recorded information. This will be addressed by excluding prescriptions with illegible handwriting or incomplete records.

Logistics challenge is a concern for CHAM health facilities which are mostly in remote areas. This will be addressed by effective planning and communication, using a combination of multiple communication channels. Additionally, the study involves managing and analysing large volume of data. This will be mitigated by thoroughly training data reporters to ensure collected data requires minimal cleaning, and by using KoboToobox to reduce most of the manual work required for data entry.

### Dissemination plan

The findings will be shared to all CHAM facilities at CHAM Annual General Meeting (AGM), the MOH AMR Sub technical working group (TWG) and Kamuzu University of Health Sciences (KUHES) Research Dissemination Conference. Additionally, the study will be presented in other local and international conferences on AMR. Finally, the study findings will be considered for publication in peer reviewed journals.

### Study amendment and termination

Study alterations will be done in full consultation with all study investigators with approval from ethical oversight committee. In the end, we will provide reports and communication to all key stakeholders in the project following termination of the study.

## Supporting information

**S1 File. Health facility sample selection using random numbers generated by Epitool.**
(PDF)

**S2 File. Data collection tools.**
(PDF)

## Acknowledgments

We thank CHAM's management for authorising the implementation of this study in the organisation's health facilities. Special thanks to the managements of the study's health facilities unwavering support during data collection.

## Author Contributions

**Conceptualization:** Evans Chimayi Chirambo, Dumisani Nkhoma, Innocent Chibwe.

**Data curation:** Francis Kachidza Chiumia, Sharon Odeo, Patience Khomani, Simon Matchado.

**Formal analysis:** Evans Chimayi Chirambo, Francis Kachidza Chiumia, Dumisani Nkhoma, Collins Mitambo, Sharon Odeo, Patience Khomani, Simon Matchado.

**Funding acquisition:** Evans Chimayi Chirambo, Elled Mwenyekonde.

**Investigation:** Evans Chimayi Chirambo, Francis Kachidza Chiumia, Innocent Chibwe, Elled Mwenyekonde.

**Methodology:** Evans Chimayi Chirambo, Francis Kachidza Chiumia, Dumisani Nkhoma, Sharon Odeo, Patience Khomani.

**Project administration:** Evans Chimayi Chirambo, Pacharo Matchere, Beatrice Chiweza, Elled Mwenyekonde.

**Resources:** Evans Chimayi Chirambo, Collins Mitambo, Innocent Chibwe, Beatrice Chiweza, Elled Mwenyekonde.

**Software:** Evans Chimayi Chirambo, Simon Matchado.

**Supervision:** Evans Chimayi Chirambo, Pacharo Matchere, Beatrice Chiweza, Elizabeth Kampira, Happy Makala.

**Validation:** Evans Chimayi Chirambo, Simon Matchado.

**Visualization:** Evans Chimayi Chirambo, Simon Matchado.

**Writing – original draft:** Evans Chimayi Chirambo.

**Writing – review & editing:** Francis Kachidza Chiumia, Dumisani Nkhoma, Collins Mitambo, Sharon Odeo, Patience Khomani, Pacharo Matchere, Elled Mwenyekonde.

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
