## [Decision Letter · Decision Letter 0]

11 Sep 2024

PONE-D-24-24627Assessing Antimicrobial Use Patterns in Christian Health Association of Malawi (CHAM) Health Facilities: A Cross-Sectional Study ProtocolPLOS ONE

Dear Dr. Chirambo,

Thank you for submitting your manuscript to PLOS ONE. After careful consideration, we feel that it has merit but does not fully meet PLOS ONE’s publication criteria as it currently stands. Therefore, we invite you to submit a revised version of the manuscript that addresses the points raised during the review process.

In addition to few minor things, this research protocol has some confusions regarding sampling procedure, which needs to be addressed at this stage. Please see the detailed comments below for further suggestions.

We look forward to receiving your revised manuscript.

Kind regards,

Muhammad Farooq Umer, PhD Epidemiology and Health Statistics

Academic Editor

PLOS ONE

Journal requirements: 1. When submitting your revision, we need you to address these additional requirements. Please ensure that your manuscript meets PLOS ONE's style requirements, including those for file naming. The PLOS ONE style templates can be found at https://journals.plos.org/plosone/s/file?id=wjVg/PLOSOne_formatting_sample_main_body.pdf and https://journals.plos.org/plosone/s/file?id=ba62/PLOSOne_formatting_sample_title_authors_affiliations.pdf. 2. We note that you have indicated that there are restrictions to data sharing for this study. For studies involving human research participant data or other sensitive data, we encourage authors to share de-identified or anonymized data. However, when data cannot be publicly shared for ethical reasons, we allow authors to make their data sets available upon request. For information on unacceptable data access restrictions, please see http://journals.plos.org/plosone/s/data-availability#loc-unacceptable-data-access-restrictions.  Before we proceed with your manuscript, please address the following prompts: a) If there are ethical or legal restrictions on sharing a de-identified data set, please explain them in detail (e.g., data contain potentially identifying or sensitive patient information, data are owned by a third-party organization, etc.) and who has imposed them (e.g., a Research Ethics Committee or Institutional Review Board, etc.). Please also provide contact information for a data access committee, ethics committee, or other institutional body to which data requests may be sent. b) If there are no restrictions, please upload the minimal anonymized data set necessary to replicate your study findings to a stable, public repository and provide us with the relevant URLs, DOIs, or accession numbers. Please see http://www.bmj.com/content/340/bmj.c181.long for guidelines on how to de-identify and prepare clinical data for publication. For a list of recommended repositories, please see https://journals.plos.org/plosone/s/recommended-repositories. You also have the option of uploading the data as Supporting Information files, but we would recommend depositing data directly to a data repository if possible. Please update your Data Availability statement in the submission form accordingly. 3. Your ethics statement should only appear in the Methods section of your manuscript. If your ethics statement is written in any section besides the Methods, please delete it from any other section.  4. Please include captions for your Supporting Information files at the end of your manuscript, and update any in-text citations to match accordingly. Please see our Supporting Information guidelines for more information: http://journals.plos.org/plosone/s/supporting-information. 

Reviewers' comments:

Reviewer's Responses to Questions

**Comments to the Author**

1. Does the manuscript provide a valid rationale for the proposed study, with clearly identified and justified research questions?

Reviewer #1: Yes

Reviewer #2: Yes

Reviewer #3: Yes

2. Is the protocol technically sound and planned in a manner that will lead to a meaningful outcome and allow testing the stated hypotheses?

Reviewer #1: Yes

Reviewer #2: Yes

Reviewer #3: Partly

3. Is the methodology feasible and described in sufficient detail to allow the work to be replicable?

Reviewer #1: Yes

Reviewer #2: Yes

Reviewer #3: No

4. Have the authors described where all data underlying the findings will be made available when the study is complete?

Reviewer #1: Yes

Reviewer #2: Yes

Reviewer #3: Yes

5. Is the manuscript presented in an intelligible fashion and written in standard English?

Reviewer #1: Yes

Reviewer #2: Yes

Reviewer #3: Yes

6. Review Comments to the Author

You may also provide optional suggestions and comments to authors that they might find helpful in planning their study.

Reviewer #1: It is a well thought out and well written research proposal on a very important topic (AMR) in a location where little is known about the problem.

The research when completed will contribute to the current knowledge on the topic.

The proposal is for a prospective study and proposed period of study is January 2024 to December 2025. However, the ethics approval for the study was only obtained on May 15, 2024. The study period therefore needs to be revised.

Typo error on line 162, the words 'Ninety percent' were repeated in the sentence.

Reviewer #2: Check minor editorial corrections on pages 14 and 16

Line 200 – are these facilities to be excluded? Please state that clearly

Line 202 – replace with (included) - otherwise the statement is not clear

Line 262 - Where the data from the multiple sources are contradictory, how will this be resolved?

Reviewer #3: Line 160: Authors should take out 'in the' as used as the last words in the line

Line 162: ninety-percent has been repeated in the same line

Authors should come clear on the sampling procedure. Is 194 the total CHAM facilities of it was sampled? If it was sampled then how did the arrive at 194? If it's the former then the sentence should be as specific as possible to clear any ambiguity. Also, if WHO recommends 3000 for the prescribing indicator, then how did authors arrive at 104 for withdrawal of consent and incomplete responses? What factor was used in this case?

Patient indicator sample size has been indicated as 274 with an excess of 113 for consent withdrawal and incomplete responses yet it is not clear how authors has authors arrived at the 113. That being said, authors went ahead to indicate that they will collect data from 2 patients per facility to make the sample size 388 which is very clear and understandable. If authors want to do 2 respondents per facility they should stick to that approach instead of complicating matters by trying to calculate a sample size using a formula with unexplained factors. I suggest authors delete the sample size calculation and rather maintain the 2 interviews per facility

Grammar check, line 218-219

Grammar check, line 223

In the safety plan, authors should provide some PPEs for data collection team to prevent risk of contamination considering they will be engaging patients on admission and reviewing records that might have travelled throughout the hospital environment

7. PLOS authors have the option to publish the peer review history of their article (what does this mean?). If published, this will include your full peer review and any attached files.

Reviewer #1: No

Reviewer #2: **Yes: **Prof. Tanimola Makanjuola Akande

Reviewer #3: **Yes: **George Akowuah

---

## [Author Response · Author response to Decision Letter 0]

28 Sep 2024

The Editor,

PLOS ONE

Dear Editor,

Re: Response to editorial and reviewer’s comments

Thank you for the editorial and reviewers’ comments which we received on 12th September 2024. We have revised the manuscript accordingly and the following are the responses: 

Section A: Editorial review and comments

Response: We have formatted the article according to PLOS ONE’s formatting guidelines including title_authors_affiliations guidelines, and main body guidelines.

Response: There are no restrictions on data availability and sharing from our side. The response in the system was because this is a protocol, and we have not collected any data yet, as we wanted to accommodate any changes, including those from the journal, for consistency and reproducibility. We will make all data from the study available and shareable. Therefore, we have updated the data availability and sharing statements accordingly.

Response: We have removed the ethical statements from the abstract and any sections other than the methods.

Response: This has been changed and formatted accordingly.

Section B: Reviewers Comments

Reviewer #1:

1. It is a well thought out and well written research proposal on a very important topic (AMR) in a location where little is known about the problem. The research when completed will contribute to the current knowledge on the topic.

Response: Thank you for the kind complements.

2. The proposal is for a prospective study and proposed period of study is January 2024 to December 2025. However, the ethics approval for the study was only obtained on May 15, 2024. The study period therefore needs to be revised.

Response: Thank you for the observation. We have revised the study period to October 2024 to June 2025. However, the Gantt chart containing tasks and timeline will retain all tasks prior to ethical approval and data collections.

3. Typo error on line 162, the words 'Ninety percent' were repeated in the sentence.

Response: We have deleted the repeated words.

Reviewer #2: 

1. Check minor editorial corrections on pages 14 and 16.

Response: Thank you for the corrections. We have adopted the suggestions.

2. Line 200 – are these facilities to be excluded? Please state that clearly

Response: We have revised this section for clarity by providing clear inclusion and exclusion criteria. We will exclude only those health facilities where services are completely shut down or not operating. It is common for CHAM facilities to close for prolonged periods due to land disputes or staff wrangles. Additionally, we will exclude health facilities that cannot be accessed by any means of transport or communication. These health facilities include those located on remote islands or in mountainous areas, where transport and phone networks are severely limited.

3. Line 202 – replace with (included) - otherwise the statement is not clear

Response: Thank you. We have revised this section for clarity by providing clear inclusion and exclusion criteria.

4. Line 262 - Where the data from the multiple sources are contradictory, how will this be resolved?

Response: Clarification from the clinicians will be sought where data from the reviewed documents are contradictory. We have revised the section to provide more clarity. 

Reviewer #3: 

1. Line 160: Authors should take out 'in the' as used as the last words in the line

Response: Thank you. This has been deleted, and the paragraph has been clearly re-written.

2. Line 162: ninety-percent has been repeated in the same line

Response: Thanking you for noting this. The repeated words have been deleted.

3. Authors should come clear on the sampling procedure. Is 194 the total CHAM facilities of it was sampled? If it was sampled then how did the arrive at 194? If it's the former then the sentence should be as specific as possible to clear any ambiguity. Also, if WHO recommends 3000 for the prescribing indicator, then how did authors arrive at 104 for withdrawal of consent and incomplete responses? What factor was used in this case?

Response: Thank you for your response. We have made significant changes to the sample sizes for the study, and this will be communicated to research ethics institution review board (S8 File). For the health facility and prescribing data sample sizes, we have used the new WHO recommendations. The recommended number of health facilities for a cross-sectional study is at least 20. To account for withdrawal of consent and incomplete data, we added 10%, resulting in a final sample size of 22 health facilities. Similarly, the recommended sample size for prescribing data is at least 600, and we added 10% for the same reasons, resulting in 660 prescriptions. We have also clearly outlined the methods for selecting health facilities and prescriptions. Our goal is to ensure a truly representative sample size and selection process, using probability and randomization methods.

4. Patient indicator sample size has been indicated as 274 with an excess of 113 for consent withdrawal and incomplete responses yet it is not clear how authors has authors arrived at the 113. That being said, authors went ahead to indicate that they will collect data from 2 patients per facility to make the sample size 388 which is very clear and understandable. If authors want to do 2 respondents per facility they should stick to that approach instead of complicating matters by trying to calculate a sample size using a formula with unexplained factors. I suggest authors delete the sample size calculation and rather maintain the 2 interviews per facility

Response: Thank you for your response. Again, we have made significant changes to the sample sizes for the study, and this has been communicated to ethical review board. The sample size for patient data, calculated using the cross-sectional study formula by Jaykaran Charan and Tamoghna Biswas. Using the formula and assumptions as put in the article, the sample size is 275 patients, and when we add 10% to account for withdraw of consent or incomplete data, we get the final sample size for patient data as 303.

5. Grammar check, line 218-219

Response: We have revised the passage accordingly.

6. Grammar check, line 223

Response: We have revised the passage accordingly 

7. In the safety plan, authors should provide some PPEs for data collection team to prevent risk of contamination considering they will be engaging patients on admission and reviewing records that might have travelled throughout the hospital environment

Response: We have included a statement regarding the provision of PPE during data collection. Data collectors will be provided with aprons, gloves, face masks, and hand sanitizers. Additionally, they will be encouraged to wear lab coats or scrub suits during data collection.

C. Our research team corrections

Team Comment: The team reviewed the entire document with a fresh perspective and addressed any typographical and grammatical issues throughout the article.

We believe that these revisions have significantly improved the quality of the paper to the satisfaction of the reviewers. 

Yours faithfully,

Evans C. Chirambo, BPharm (Hons), Msc. CPP, Pre-PhD Fellow.

---

## [Decision Letter · Decision Letter 1]

27 Nov 2024

Assessing Antimicrobial Use Patterns in Christian Health Association of Malawi (CHAM) Health Facilities: A Cross-Sectional Study Protocol

PONE-D-24-24627R1

Dear Dr. Chirambo,

We’re pleased to inform you that your manuscript has been judged scientifically suitable for publication and will be formally accepted for publication once it meets all outstanding technical requirements.

Kind regards,

Muhammad Farooq Umer, PhD Epidemiology and Health Statistics

Academic Editor

PLOS ONE

Additional Editor Comments (optional):

Reviewers' comments:

Reviewer's Responses to Questions

**Comments to the Author**

1. Does the manuscript provide a valid rationale for the proposed study, with clearly identified and justified research questions?

Reviewer #3: Yes

2. Is the protocol technically sound and planned in a manner that will lead to a meaningful outcome and allow testing the stated hypotheses?

Reviewer #3: Yes

3. Is the methodology feasible and described in sufficient detail to allow the work to be replicable?

Reviewer #3: Yes

4. Have the authors described where all data underlying the findings will be made available when the study is complete?

Reviewer #3: Yes

5. Is the manuscript presented in an intelligible fashion and written in standard English?

Reviewer #3: Yes

6. Review Comments to the Author

You may also provide optional suggestions and comments to authors that they might find helpful in planning their study.

Reviewer #3: The manuscript has indeed undergone a major transformation since the last review. I am very impressed with the level of detail and clarity in the current manuscript.

7. PLOS authors have the option to publish the peer review history of their article (what does this mean?). If published, this will include your full peer review and any attached files.

Reviewer #3: **Yes: **George Akowuah

---

## [Editor Report · Acceptance letter]

5 Dec 2024

PONE-D-24-24627R1 

PLOS ONE

Dear Dr. Chirambo, 

I'm pleased to inform you that your manuscript has been deemed suitable for publication in PLOS ONE. Congratulations! Your manuscript is now being handed over to our production team.

Kind regards, 

on behalf of

Dr. Muhammad Farooq Umer 

Academic Editor

PLOS ONE